# Street mothers' well-being and motivation to leave street life in Bahir Dar city, Ethiopia: A phenomenological study

**Dabere Nigatu**[1]*, **Gebeyehu Tsega**[2], **Shiferaw Birhanu**[3], **Yinager Workineh**[3], **Christian Tadele**[4], **Fentie Ambaw**[4]

**1** Department of Reproductive Health and Population Studies, School of Public Health, College of Medicine and Health Sciences, Bahir Dar University, Bahir Dar, Ethiopia, **2** Department of Health Systems Management and Health Economics, School of Public Health, College of Medicine and Health Sciences, Bahir Dar University, Bahir Dar, Ethiopia, **3** Department of Pediatrics and Child Health Nursing, School of Health Sciences, College of Medicine and Health Sciences, Bahir Dar University, Bahir Dar, Ethiopia, **4** School of Public Health, Bahir Dar University, Bahir Dar, Ethiopia

* daberen@yahoo.com

## Abstract

### Background

Being-street mother is a challenging life situation for both the mothers and their children. However, the lived experiences of motherhood in street families are not explored very well in Ethiopia in general. Hence, this study explored street mothers' well-being, perception of street life, and motivation to leave street life in Bahir Dar city, Ethiopia.

### Methods

A phenomenological study was conducted on 10 street mothers from July 13, 2021 to July 17, 2021. The mothers were selected using purposive sampling technique. Data were collected using face-to-face in-depth interview method. Data were analyzed using framework approach.

### Results

Four themes emerged from the data: well-being of mothers and their children with four sub-themes (physical, social, mental, and spiritual wellbeing), perception of street life, motivations to leave street life and efforts to end street life. Nearly all of the street mothers perceived that living on the street was terrible for them and their kids. They described it as an absolutely revolting, bitter, awful, horrible, and difficult life. Generally, street mothers had the motivation to leave street life, but only some had exerted tangible efforts to end the street life.

### Conclusion

Street mothers had a very poor status in almost all dimensions of well-being. The perception of mother about their street life was negative. The mothers had strong motivation to end

**Data Availability Statement:** All relevant data are within the manuscript and its Supporting Information files.

**Funding:** The author(s) received no specific funding for this work.

**Competing interests:** The authors have declared that no competing interests exist.

street life but were unable to make strong tangible efforts showing that they need assistance mechanisms before they change to street extended families under misery.

## Introduction

Well-being, perception and motivation are interconnected human elements [1]. Moral philosophers discuss well-being as the most basic elements of good lives [2]. In its crude meaning, well-being refers to contentment, satisfaction, or happiness derived from optimal functioning [3]. Martino 2017 defines well-being in terms of objective, external and universal notions of quality-of-life indicators such as social attributes (health, education, social networks and connections) and material resources (income, food and housing) [4]. It can also be defined as multidimensional entity that composed of physical, mental, social, emotional, economical and spiritual dimensions [5–7]. According to the World Health Organization (WHO) proposed definition, well-being exists in two dimensions: subjective and objective. It comprises an individual's experience of their life and a comparison of life circumstances with social norms and values [3, 6, 7]. The people's subjective well-being involves affective and cognitive evaluations of their own lives [6–9]. Despite the varying definition of well-being, it has got high attention internationally and nationally [10, 11]. However, well-being of the people is not maintained, especially in economically disadvantaged segment of population. Street families are among the economically disadvantaged population [12].

Well-being and perception are linked. Perception is an interpretation of the primitive intuition, an interpretation apparently immediate, but in reality, gained from habit corrected by reasoning [13]. The main task of human perception is to amplify and strengthen sensory inputs to be able to perceive, orientate and act very quickly, specifically and efficiently [13, 14]. For instance, perceived quality of life refers to how people perceive and evaluate their life. It is a perception that reveals the subjective evaluation of the life experience. The components focus on overall life satisfaction and happiness, as well as satisfaction with specific domains of life, e.g., marriage, interpersonal relationship, work, leisure activities, and health [9]. People's self-perceptions about their health are very important in the present as health outcomes [15]. Thus, knowledge about well-being is gained through people's own perceptions [6]. Except one study we have accessed [16], perception about street life or street people is mostly studied from the perspectives of non-street people while the perspectives of street people towards street life is overlooked.

In Ethiopia, most people live in multidimensional poverty lacking access to at least three of the fundamental needs such as adequate nutrition, health, and shelter [17]. Evidences show the link between poverty and streetism in Ethiopia: multidimensional poverty is a predisposing factor for streetism [18, 19], which refers to the life situations of street people who usually live in the streets and engage in menial income [20]. Street families are formed because of various factors. The factors are often related to economic and social disruption including, poverty; dysfunctional families; political unrest; domestic violence; human trafficking; mental health problems; substance abuse; and unemployment [21, 22]. In the Ethiopian context, Heinonen noted that family circumstances to be the major cause that spurs many children to streetism [23].

For children, several agents of socialization exist across settings. Family setting is certainly the most important agent of socialization for infants and young children [24]. Family is where the vast majority of people learn the fundamental skills for life. Parents' values and behavior patterns profoundly influence those of their daughters and sons [24, 25]. However, in addition

to poverty, the ability of parents of street children to provide appropriate care for their children were hindered by existing home and street related physical environments [23].

In fact, even in nations with robust census or civil registration systems, getting reliable and valid data on the number of homeless or street people in a given area or country could be a challenging phenomenon [26]. Enumeration of homeless individual is difficult because they typically seek to conceal themselves from the authorities, so they intentionally hide from enumerators. Additionally, the geographical extent to cover may encompass all the legal jurisdiction of a given country since they lack fixed residential place [26, 27]. In Ethiopia also the lack of accurate estimate on the magnitude of street people is a persisting problem [28]. Despite this fact, there are some qualitative and quantitative evidences indicating that the number of individuals living on the streets has increased over time in many towns/cities of the country. According to the 2007 census report, 5,210 homeless people exist in Addis Ababa and 25,020 across the entire county [28, 29]. Whereas in 2018, the Ministry of Labour and Social Affairs reported that around 24,000 homeless people exist in Addis Ababa; approximately 10,500 street children and 13,500 homeless adults [28]. A study done in Bahir Dar city noted that adult women account one-third of the homeless population. Many of them came directly from rural settings. The prime pulling factors to have many street people in Bahir Dar city were because of being a promising city for job, being a state capital, big tourist attraction and big religious center for the Orthodox Christians [30]. The changes in the family dynamics is also among the most significant factors in the growth of streetism [20].

For mothers with children, the lack of secure accommodation or creche facilities cannot allow them to move far from home to look for income opportunities [28, 31]. Hence, they were forced to live in rented houses, slums and along the roads where civic amenities were either absent or measurable [31]. Generally, family structures and environments have contributed for the persistence of poverty and inequality, and differences in children's educational outcomes [32].

There are studies addressing the push and pull factors for children and adults of both sex to streetism [16, 18, 21, 22, 28, 30, 31, 33, 34]. However, the well-being of street mothers and their children, mothers' perception about street life and motivation to end street life are not explored very well in Ethiopia in general and in Bahir Dar city in particular. Studying these issues are important to inform policy-makers, programmers and implementers working to improve the life situations of street mothers and children. Hence, the current study was aimed to explore the well-being of street mothers and their children, street mothers' perception about street life, and motivation and efforts to end street life in Bahir Dar city.

## Methods

### Study design

A phenomenological approach was employed to study lived experiences of street mothers. The study was conducted from July 13 to July 17, 2021.

The phenomenological study design advocates the study of direct experience taken at face value. It sees behavior as determined by the phenomena of experience. Phenomenologists agreed on core philosophical viewpoints as a belief that the consciousness is central and understanding the subjective consciousness is important. Description of events as they appear as a method of knowing in phenomenology is fundamental because it is a matter of describing, not of explaining or analyzing [35]. Phenomenological research is a strategy of inquiry in which the researcher identifies the essence of human experiences about a phenomenon as described by participants. Understanding the lived experiences marks phenomenology as a philosophy as well as a method, and the procedure involves studying a small number of subjects through

extensive and prolonged engagement to develop patterns and relationships of meaning [36]. Through close examination of individual experiences, phenomenological analysts seek to capture the meaning and common features, or essences, of an experience or event [37]. Thus, since this study aimed to explore the lived experience of street mothers, we have used a phenomenological approach.

## Study setting

The study was conducted in Bahir Dar city, Ethiopia. Bahir Dar is the capital city of Amhara regional state. The city is found on 1801m above sea level and is located 564km North West from the capital city of Ethiopia. The city has 17 Kebeles, three governmental and four private hospitals. There are many market areas, buildings, asphalt and cobblestone roads, churches and mosques in the city. Besides their primary purpose, these infrastructures and religious institutions are serving as the means of survival for street individuals. The investigator team did not access any recently published evidence showing the prevalence of street people in Bahir Dar city. However, based on our (investigators' team) observation there are many street people in the city and it is on raise overtime due to the current internal conflict in the surrounding districts in Amhara region and other reasons. Dube 2014 noted that the city had a high rate of street people directly fled from rural setting since it was a promising city for employment, the state capital, a popular tourist destination, and a significant religious center for Orthodox Christians [30].

## Population and sampling strategy

The study participants were mothers who have at-least one child and who were on the street. We interviewed 10 street mothers who have at least one child. The sample size was determined based on information saturation. Three criteria were used to decide on information saturation. These include; checking of the research questions for adequately answered, the answers become complete, and no new information comes from interviewees. After each interview, a discussion was also held with the investigators team.

The study participants were selected by purposive sampling technique. Mothers were selected from different age groups and duration of living on street in-order to obtain deep information about lived experiences of street mothers. First, we scanned the city for mapping of potential sites. During the scanning/mapping phase, we travelled on the main roads in the city to identify and have a mental map of the potential sites, where we could access street mothers for interview. Second, we teamed up into two groups and visited the potential sites and interviewed the mothers. In addition to the main road sites, we visited the church areas since they are potential sites to access street mothers. As the data collection progress, each day, we had a common session to discuss participants mix by age and duration of stay on street. The discussion continued until we had a minimum adequate sample size and had reached the point of information saturation.

## Data collection method and procedures

Data were collected by a face-to-face in-depth interview method. The researchers developed and used an in-depth interview guide to interview street mothers. The interview guide was originally prepared in English language and then translated to the Amharic version for ease of understanding. The interview guide contains 10 main questions and complemented with probing questions. A summary of the main in-depth interview guiding questions is given in table form (Table 1). For interested reader, full version of the in-depth interview guide is appended (S1 Appendix). The researchers were teamed into two groups to interview; one

**Table 1. The main in-depth interview guide questions are organized by main themes and subthemes.**

| Main theme | Subthemes | Guiding questions and probes |
|---|---|---|
| Wellbeing | Physical wellbeing | Please explain your current physical wellbeing? (Probe: your physical wellbeing, your child's physical status) |
| | Social wellbeing | Please describe your relationship with friends, families (partner), society? (Probe: your relationship, your child's relationship) |
| | Mental or psychological or emotional well-being | How do you describe your current mental or psychological or emotional well-being? (Probe: your mental, psychological or emotional wellbeing, your child's mental or psychological wellbeing) |
| | Spiritual wellbeing | How would you describe your current spiritual well-being? (Probe: your spiritual wellbeing, spiritual practice, your child's spiritual practice) |
| | Economic wellbeing | How do you describe life on the street? (Probe: street status, economic status, income sources, life satisfaction) |
| Perception to street life | | a) How do you describe life on the street? (Probe: street status, economic status, income sources, life satisfaction) b) How do you explain motherhood life on street? (Probe: self-care (feeding; hygiene, clothes, health care), child/ren care (looking after; feeding; hygiene, clothes, health care, education) |
| Motivation to leave street life | | What do you think to leave street life? (Probe: desire, motivation to end street life) |
| Efforts to end street life | | a) What efforts you have tried to end street life? (Probe: for you, your child/ren, give example) b) Describe the supports that you need to end street life? (Probe: from gov't, community, NGO, private organization, religious organization) |

interviewer and one assistant for detailed note taking. Data collection was started with a broad and general question followed by probing questions. Then the questions get more focused as the data collection progresses. The participants were engaged during interview; first the interviewer asked the participant and listened attentively until completing their idea followed by probing questions depending on the response of the participant. The interviews were conducted in suitable place to prevent unnecessary interruptions and increase interviewees' concentration. At the beginning of the interview, the researchers introduced themself and gave enough information about the objective of the study. The interview was started when the participant feels comfortable and by taking their background information. Each interview was audio recorded using digital voice recorder. The audio recording was done after getting consent from each participant. A team of researcher interviewed one mothers per day. The average duration of the interviews was 24 minutes.

## Trustworthiness

Validity of data was assured via thorough, careful and honest questioning, and rapport building with participants. Summarization and reflection of participants' narratives were used to verify accuracy of responses. Audio recorded data were repeatedly listened, transcribed and translated by two researchers. Interpretive validity was maintained by the use of participants' own words and peer review. The investigator team carefully and repeatedly read the transcribed data until they fully understood the content or core message of the responses. The audio recorded data were triangulated with field notes, maintenance of audit trails, peer debriefing and providing rich quotes were also done to maintain credibility.

### Researchers' position

All the researchers have no relationship with the current phenomena of study, i.e., street life. All the researchers have health background and a minimum of master degree in health field with varying level of qualitative study experience. The last author, Professor Fentie Ambaw, has supervised the overall research process. All of the remaining authors were PhD students in Public Health.

### Data analysis

Immediately following each interview, the researchers wrote preliminary notes. These include; relevant facts about the interview, the setting, and the participant's condition or behavior, as well as the researcher's impressions. At the end of each day's interview, verbatim transcription was conducted with the original language of interview (i.e., Amharic version). The transcription was made by the one who interviewed the mother. Then, the researchers held general discussion on the meaning of some local language and the way forward about translation. We agreed to identify and report difficult language encountered in the translation process on each next team meeting. The translation of the transcript from Amharic to English was also done by the one who interviewed the mother. We used conceptual translation after repeated read and review of the transcribed tapes to ensure accuracy of wording and authentic representation of the participant's experience. The research team made two days review discussion on the translated transcript displaying by LCD projector.

A framework analysis method [38] was used to analyze and present the data. The WHO definition was used to guide the study and organize the sub-themes of well-being framework [6]. The summary of the themes, subthemes and codes used is given as S2 Appendix. The researchers read the transcript several times to be familiar with the data. Data coding was done line by line. Based on the relation of the codes, sub-themes were developed. Then, sub-themes were organized under main themes. Data reduction was conducted. Finally, the results were presented under the themes and subthemes. Open code 4.03 software was used to analyze the data.

### Ethical considerations

Ethical clearance letter was obtained from the Bahir Dar University College of Medicine and Health Sciences Institutional Review Board. Support letters were obtained from the School of Public Health Bahir Dar University, and Labor and Social Affairs of Amhara regional state. Besides, street mothers were informed about the purpose and benefit of the study along with their right to refuse. The interviewers read the information sheet and consent form for each study participant until they comprehend the contents. Then, study participants expressed their agreement or disagreement verbally instead of providing hand signed approval. Additionally, each study participant provided their consent to audio-record the interview. We assigned code for each interviewed mother to maintain anonymity and ensure confidentiality of information.

## Results

### Background characteristics

The age of street mothers participated in the study ranges from 20 to 42 years. Six of the mothers had no education. Almost all mothers had detached/disconnected marital relationship. Nearly, half of the street mothers involved in the study lived on street for more than a year. The street mothers had a minimum of 1 and a maximum of 4 children. Most of interviewed street mothers were used to live in rural area before joining street life.

## Well-being of street mothers and their children

The well-being of street mothers was organized in six sub-themes: physical, social, mental, emotional, economical, and spiritual sub-themes.

**Sub-theme 1: Physical well-being.** Most street mothers expressed their physical problems in-terms of physical signs and symptoms such as non-specified pains, feeling numbness, waist stiffness and backpain. A 30 years old street mother explained:

> *"In the left side, starting from head to toes [pointing to the left toe], I do not feel it as it is mine [feeling numbness]. The characteristics of the illness varies. Now, it [the illness] stiffened my leg and I feel troubling pain. Above my waist it catches my abdomen (she is pointing to her abdomen). I feel back pain. And when I feel severe back pain, I feel stiffened waist. My abdomen gets distended. Then, my urine color changes."*

Street mothers also stated as their children have physical problem. For example, a 25 years old street mother said "*My son is sick. If I am lucky, he will grow up for me, if not, I cannot afford to do anything to cure from his illness.*"

Besides, a 35 years old street mother explained her three years old child physical well-being:

> *"His eyes do not see; cannot you see? No problem, you can see him [she uncovered the baby while he is on her back. Both eyes cannot see; eye balls are not present]. Off course; both eyes are not present at birth. He cannot maintain upright position. He is just a blind that he cannot stand and walk, that is all."*

**Sub-theme 2: Social well-being.** The street mothers used to express their social well-being in respect to their relationship with families, friends, neighbors, and the community at large.

Most street mothers had bad relationship with their family while some others had good family attachment. There were also mothers who had no connection with their family.

For instance, a 40 years old street mother expressed her good relationship with her children:

> *"They [Her children] are attached to me to get motherhood affection because I love my mother too. . . . My mother was also a good person but she died. It is too hard to them to leave me here [street]. My brother is also living with me because he found hard to leave me here."*

On the other hand, many street mothers had bad relationships with families. In this regard, a 22 years old street mother stated as: "*Although he [her husband] believed that she [her child] is his daughter, he was not comfortable. My husband and relatives do not know that I am begging here [Bahir Dar city].*"

Another street mother added that:

> *"After I gave birth to this baby, they [her families] hated and stigmatized me. When my sister gave birth, they did not invite me in her child baptizing ceremony. Even, they did not call and say "how is your baby?""*

A 35 years old street mother also expressed her experience:

> *"We [respondent and her family] only talk on the phone because I do not feel comfortable to go there [her family residential place]. You know why? I do not feel comfortable living in the countryside/rural area."*

Another 40 years old street mother stated her bad relation with her family:

*"My oldest daughter is huge, and she beats/hit me hardly. She strangled my throat. She sleeps on street. . . . She quit her schooling by saying "learning does not help me." Then, I told to the police, and the police said that "if she is beyond your control, you can't do anything for her.""*

On the other hand, some street mothers had no connection with families. A 25 years old street woman remarked that *"I have no mother or father, they died, I have no brother or sister. I have only him [her child]. I have no friends. I have nothing. I just fear for my life."*

A 40 years old street mother also described that:

*"I have no connection and I have nothing to connect with my family: Only my children and God and this wall we see [showing their Kenda's house one-side wall fence that is made of bricks]. This is the most important thing when the wind blows it protects us, and this is the reward of God's help."*

Most street mothers had good relationship with their friends while some others had bad relationships. For instance, a 40 years old street mother expressed her good relation with friends:

*"If I have, we will share. I live in a rented house and she is of-street mother (she is pointing to her friend). During rainy season, I always insist her to come and spend the night with me until the rain stop. . . . If I have no food for my children, she will feed for me. If she had no food for her baby, I will feed for her. [. . .]"*

Another 40 years old street mother explained her good attachment with friends:

*"It [friend relationship] is good. They say to us "don't worry" something like that. We help each other with friends; we make coffee and have shared eating."*

A 20 years old street mother also stated the presence of good relationship with her friends:

*"We street individuals, if one has no food, the other feeds him and vice versa. For example, if the woman here has no money, I will give it to her. Or they [children] may eat by sharing. . . . we are happy to share it."*

Among the mothers who had bad relationship with friends, a 20 years old street mother quoted that:

*"Here [on street], also, my friends discriminate me. They said "why you gave birth from him [her husband]?" Even, they bought drug to terminate my pregnancy, but I did not do that."*

The community/societal relation of the street mothers was also explained as either positive or negative relations. The negative relation with the community was quoted by a 20 years old street mother:

*"We do not sleep at night. When we sleep around Dashen Bank, the guard does not allow us to sleep. We also try to sleep near to Bogale furniture, but they didn't let us sleep. The police themselves are chasing us at night by asking us "why not you rent a house?" . . ."*

A 20 years old street mother also explained the bad relationship she experienced with the society:

*"There are people who spit on us when we eat leftover food (bulie). If the government does something to me, I will leave street life because the people see us as dirty material. They belittle us when they pass here. No one from many of them see us in a safe way. They do nothing for us. So, I think if the government does something for us, we will leave."*

Furthermore, a 35 years old street mother expressed her social relation:

*"Our relationship . . ., conflict is inevitable among living things even inanimate objects can do when they come together. We, who have child/ren, also quarrel each other. Then we immediately talk peacefully."*

On the other hand, some street mothers had supportive relationship with the community. For instance, a 40 years old street mother articulated:

*"Neighbors are supportive to me, they mend when my tend get torn, reassure me not to worry. They are the one who are embracing us while my relative left me homeless (Uh). It is the Kebele 14 people who calms and reach for me when they hear any alerting sound. . . .. Though they are receptive to me they have no the capacity to do that in full, this time things are becoming worse."*

A 40 years old street mother also verbalized the supportive relationship of the community:

*"When it [the plastic cover] torn, they advise me by saying do it this way, they gave me that wood, and they gave also me this, and that one (pointing to the things that the neighbor helped her)."*

**Sub-theme 3: Mental or emotional well-being.** The street mothers' mental or emotional well-being was characterized by fear, worry, stress, and anxiety. Almost all the street mothers' had fear about child traffic accident, child theft, violence and child's future life.
For instance, a 40 years old street mother expressed her children's worry:

*"Their [children's] mental is getting tired of overthinking. On top of this, . . . they think that these all are happening to them because of the death of their father and their mother being alone. They started to tear and to talk alone. The next is being mad, (the respondent was in grief mood), they are healthy other than this."*

The stressful life of the street mothers was also expressed by a 22 years old street mother:

*"I get very stressed when I sit alone and think of street life, but while I talk about that thing [stressor] with my friends (individuals like me) or with sleeping floor renters, I relieve from it. I am very worried about where I used to be (status of me when in my house) before and where I am now, but when I talk about it, I get relieved from my worry."*

The anxiety of street mothers was stated by 20 years old mother:

*"Sometimes, you envision to reach somewhere. But, they [other people] say, "you will not improve your life by yourself" Then anxiety will come. There is a woman, who says, "why do*

*not you work as housemaid? And again, she says, "you don't change your life by yourself. However, if you become a housemaid, you will steal and comeback." At that moment, my mind gets hearted."*

A 30 years old street mother expressed the fear she had about child car accident:

*"My immature children are here on the road; they are exposed to road traffic accident when they play on the road in the whole 24 hours. I just always suspect that the cars might crush them when they move here and there in the whole day."*

Similarly, a 40 years old street mother, expressed the emotion:

*"As you are here, the madman comes, the wind comes, and beast comes, when I takeoff this (pointing towards the plastic) the rain gets the children. So, it is very difficult (tear is coming in her eyes) . . . (shedding tears on her cheeks) . . .The things that make me teared and saddened when I see my children here sheltered at me and they [stepmother and her father] left us in an open field. Ehh, ehh. . .when it gets rain, they wear a plastic."*

A 25 years old mother also expressed her emotional well-being:

*"What can I take with me? What can I eat? What shall I go out to eat? But when I am in trouble, I kept silent; and continue this street life."*

**Sub-theme 4: Economic well-being.** Street mothers' economic well-being was explained by inability to fulfill basic needs and insecure income. The street mothers underlined that food, medical care, personal hygiene materials, clothing, and housing imposed huge economic burden on their livelihood. The degree of economic well-being was inconsistent across the street mothers.

Almost all street mothers expressed their grievances about unexpectedly high economic burden of buying food. Forty years old street mother who had four children, expressed the situation:

*"We wait up to three o'clock in the evening here [on street]. Some people come and buy macaroni, something to me and go. I cook that and give it to them [children], but my stomach is empty. We are fasting, we have nothing to do."*

A 42 years old street mother also shared the burden of food shortage: *"Food! as you saw us, we sit with other beggars, while they eat daily bolus of food, we ask them to share their food."*

Children's medical care was a big challenge for street mothers unless they get some generous people to cover for them. A 30 years old street mother clearly indicated the economic burden of health care:

*"When children get sick, I cry to those people I assume rich persons who pass here [street], they do not leave me alone and they give me money for medical care when I beg them [rich persons]. If they [children] are sick, they give money for medical care to them, and they [rich persons] never leave us."*

Some street mothers expressed the high economic burden of fulfilling personal hygiene materials. A 22 years old street mother expressed as:

*"I buy a jar (jerry-can) of water by one birr and I use it. I use that jar of water carefully, not to frequently buy or pay another one birr."*

The economic well-being of street families related to clothing and housing was also very critical almost in all street mothers. This was explained by a 25 years old street mother as follow:

*"We live at holy water house (tsebel bet), eight to ten individuals in a room. Sometimes, in summer, the number decreases. I do not even have clothes for me and for my child. Someone gave us a blanket (koborta) and we wear it while we sleep."*

The average daily income among some street mothers was ranged from 30–100 birrs. This level of income was explained by a 35 years old street mother:

*"I can get 50birr or 60birr per day and sometimes I can get also 100birr. The money I get varies from day to day."*

**Sub-theme 5: Spiritual well-being.** The spiritual well-being of the street mothers was expressed by praying, thanking God, attending holy water, taking flesh-blood and joining monasteries. According to Ethiopian Orthodox Christian faith, there are holy water sites known for healing miracles. The Orthodox Christian followers travel for miles or even days to seek the curative powers of holy water blessed by Orthodox priests. They drink holy water and/or submerge themselves in pools to receive healing. The frequency and duration of such practice can be determined by the managing priest and may vary from follower to follower. Overall, Orthodox Christianity followers who are attending holy water sites believe that holy water has curative power and can heal any illness [39].

A 30 years old street mother explained her religious attachment:

*"I got sick five years ago. As it started, I went to Gishen Debre Kerbe holy water. And I also went to St. Michael and St. Abunehara Dingel holy water. Then I got improved." The same mother also remarked that: "I sat in St. George cave for eight months. I was baptized for eight months by eating only dry maize."*

A 35 years old street mother also voiced her religious attachment:

*"I buy it [necktie] from the sellers located on the roadside and tie on their neck like this (she is pointing the necktie on her elder child). Except this, even I haven't tried to tie an evil medicine because I believe God protect them [children]".* She demonstrated her prayer by saying *"God be with us; you know what you gave me." "Since He is the one who gave me my children and He is the one who can take them when He wants."*

A 30 years old mother stated that

*"I have nothing. My idea is, if I don't recover [from her illness], with this age, by the age of 30, I beg, I plan to take over this big child [middle child] to rural village and I turn and join to a monastery with a little child. Where will I go with this disability forever? If I get a better monastery, I will take His flesh-blood (sigawo demu)."*

A 35 years old street mother also verbalized the religious practice:

*"I always pray my God. During sleep time, I always tell Him [God] that please give me what I deserve and then I can take my own choice. Always, while I am sleeping, I am informing my God to quit from this type of work. "Now, I and He [God] know how He gives it [street life] for me. Unwise man thinks that it [being a street mother] is by my interest. But there is nothing such a disgusting thing. In the name of the Father, and of the Son and of the Holly Spirit. . ."*

A 40 years old street mother also described her spirituality:

*"So far, I survived here because of the will of God. I wake up at night and say this is a challenge what the God give for whom He loves them. This is the word of God which gives me the strength. And I can't do anything. I wish God to bring a better/blessed day for us and find a solution for my children."*

A 35 years old street mother expressed her religious thinking: *". . .These children; first this child's [little child] disability that God make him lost his body. So, I came here not to let him die. And so, I came and so far, I am trying to survive, thanks to Him."*

## Perception of street life

Almost all of the street mothers perceived street life as very bad to them and their children. They expressed it as very disgusting, bitter, bad, horrible, hard life, and not a life at all. The street mothers have developed such perceptions because of their lived life challenges they faced while they live on street. The lived life challenges include cold weather and rain during summer, excessive sunlight exposure during the dry season, lack of sleeping place at night, insults, and difficulty of safeguarding their children and satisfying other basic needs. In crude terms, the street mothers underlined that the street life was very hard for girls/mothers than men.

The following street mothers' quotes were clearly indicating these facts:

A 40 years old street mother expressed:

*"It is a matter of loss. Living on the street is not a life at all. For me, it is not a job either. Now, for example, we are looking at the face of others whether they give something to us or not. We have no choice, with children, where can we go? But I, myself, so overwhelmed. Selling maize here [on street] is an alternative work instead of sitting to beg here. I prefer this (selling maize), I don't want the street life."*

Another, 40 years old street mothers remarked the suffering they endured in this way:

*"Uh. . .how this could be a life at all, during day time the sun, and at night time also we sleep under plastic shelter "Kenda". We are living here [on street] because a man cannot be buried alive. I am suffering and my children are also suffering, nothing is attractive as you live here."*

A 20 years old street mother also stated the ugly face of motherhood on street:

*"Motherhood is very difficult. Um, I think the males do not care about their children. Even if my husband is not here, of course, I observe others father here around. And a girl suffers a lot. Breastfeeding is very painful since we [street mothers] have no food to eat. And street life is very difficult for a woman, especially for a mother."*

Similarly, a 25 years old street mother said:

"*There are insults, some people say the beggar. But I can handle it by being patience. There are so many types of insults. Street life needs being patience. It is hard to say. . . .If I find him [her husband], I will leave the street life. The street work itself is very disgusting. It is a shame. It is an insult for the family too.*"

A 35 years old street mother also remarked the horrible feature of street life:

"*Street life is bad, but what is good? It is horrible to see a human face, one swears in words and the other says to me "where will she reach with these children? what is her life direction?" Some people say "she cannot work with these children in the light of her life." In general, for me, it is just a bad task. I think. . . It is horrible.*"

There were mothers who perceived their children are at risk of traffic accidents and being stolen by someone while living on street. Street mothers spent long time on street which is unsafe place for their children because children's demand safe place and adult protection while playing, feeding and do other things.

A 30 years old street mother stated the fear she had about the risk of car accident her children endured:

"*I have never felt so overwhelmed. I live always with unrelieved worry. When they [her children] go there (pointing to the road), I say, "Stop!" while cars pass via two opposite directions. I do not have a leg to run and bring them, I just always suspect that the cars crush them. They don't know when you tell them since they are babies because they do not know the bad and good things. . ..*"

A 22 years old street mother expressed her perception about the risk of child theft:

"*I am afraid of my child being stolen since she approaches with others by calling my father and my mother. I am afraid she will be crushed by a car. There is no money for medical care.*"

Some street mothers were also perceived that they were considered as normal by other people though they had a health problem. In fact, some mothers actually received bad judgments from people while some others actually didn't but they perceived it in such a way.

A 30 years old street mother expressed:

"*When people see me above my waist, they perceive me as normal (she is pointing the waist), and they look me in amazement while I beg.*"

A 35 years old street mother also expressed:

"*Now, I and He [God] know how He gives it [street life] for me. Unwise man thinks that it [being a street mother]is by my interest. But there is nothing such a disgusting thing. In the name of the Father, and of the Son and of the Holly Spirit. . .*"

On the other hand, few mothers perceived the street life as it was not that much a problem and even better than tiresome works such as washing clothes and other things. A 42 years old street mother described street life:

"*. . .I am thinking nothing; I am just saying the street life is better. We are new. Street life is not that much worse, it is somewhat good. It is better than being tired from washing clothes and other things.*"

## Motivation to leave street life

Almost all street mothers had desire to leave the street life. The motivational factors to leave the street life were the intent to improve their children life or their own life or both. The desire to educate their children, to invest on their children, and envisioning better future work for their children were the motivational derive for improved children life. Whereas the motivational factors that intended to improve their own and their children's life were desire to: saving money, work with permanent salary/income, have their own house, self-contained life, and refrain from blame, insult and discrimination.

Most street mothers had desire to leave the street life and educate/invest on their children, at least to make their children's life out of the street in the future. This is expressed by a 30 years old street mother:

*"I believe that this child (she is pointing to the older child) should join school, but I do not know the place [school]. And also, when he comes back from school, he will help me to fetch water."*

Similarly, a 20 years old street mother described that:

*"If I get a job in the future; I will like to work, rent dormitory and want to teach my child in kindergarten. If someone helps me a little, I will use my maximum effort. If I get a school that teaches my child the whole day, I want to work."*

The same mother remarked that:

*"If the government does something to me, I will leave [street life] because the people see us as dirty material. They belittle us when they pass here. No one from many of them see us in a safe way. They do nothing for us. So, I think if the government does something for us, we will leave."*

A 22 years old street mother added:

*"I will teach my child, and I want to be self-contained rather than being here the whole day. If I get a little work, if God wills, I will expand it [work]. A little work is better than sitting in the street throughout the day."*

Moreover, a 35 years old street mother described that

*"If they [her children] grow-up, I get money and container; I can do different things because I can write and speak. I do not live like everyone else here [street]. When they [kids] know what is right and what is wrong and if I find something, I can do."*

Some street mothers had desire to leave street life due to pressure from their children. For example, a 40 years old street mother described that *"They [her children] are insisting me to let get out of here and go to a rural town if I can't afford living here otherwise. They just sit here and do their daily homework saying "what are we going to eat here?"*

## Efforts to end street life

Some street mothers have tried a variety of methods to end their children's lives on street, including teaching them, giving them to relatives or others, and allowing them to focus their time on job and education.

For example, a 30 years old street mother who gave her child to a person explained:

*"The child [oldest child] is more mature than the other children; I gave her to someone else. Michael knows, I am always crying in front of Michael to get a way get out from this life."*

Another mother described the efforts she made to educate her child and the challenges she endured in the process:

"*My oldest child was grade seventh student here in Sertse [the name of the school]. She [her child] said that "learning would not help me." I told the police, "He said that if she is beyond your control, you can't do anything for her." Instead of this, I raise these [the two children] by begging to the best of my ability. This one (she is pointing to the middle child) is in school. She (the mother is pointing to little child) is too young to start school. I tried much too, but they refuse me to enroll due to her age"*.

A 20 years old street mother also described her effort to generate income:

*"I wanted to leave street life before, but I failed to do so. And someone promised for me to start shoeshine. But he did not. Then, I started shoeshine by myself. But I stopped it because there is no income to sustain it. I think we will leave if we do something else."*

The findings suggested that there was clear interaction between themes and sub-themes. Thus, we depicted a model showing the relationship between each theme of the overall lived experiences of street mothers (Fig 1).

## Discussion

Evidence generated over the last two decades shows that disadvantaged social circumstances are associated with increased health risks [40]. The current study explained street mothers' and children's well-being by physical, social, mental or emotional, economical and spiritual domains. Actually, well-being is a multi-dimensional construct [5]. The proximal drivers of well-being are linked to broader social conditions, such as low and insecure income, poor housing and working conditions, inadequate transportation systems, and misguided agricultural and education policies [41]. Noting that the Dodge et al defined well-being to center on a state of equilibrium or balance that can be affected by life events or challenges. Further, the paper remarked that stable well-being is when individuals have the psychological, social and physical resources they need to meet a particular psychological, social and/or physical challenge [42]. The current study explored various forms of physical limitations and physical complaints as expressed by street mothers. The physical complaints include feeling numbness, unspecified pain, backpain, troubling pain, hip joint stiffness, swelling, abdominal distention, unable to stand and walk, and blindness. The physical well-being is linked with other well-being domains. There is evidence that people who report ease of communication with their parents are more likely to report fewer physical complaints. They are also less likely to participate in aggressive behaviors [40].

The current study explored that the social well-being of street mothers was predominantly explained in terms of disconnected relationship. The social characteristics demonstrated in their interaction with families, friends, and society. Almost all studied street mothers had disconnected marital relationship (widow, divorced or separated spousal relationship). A study done in Nigeria stated that street people are considered as most marginalized and socially excluded section of the society. Their close relationships with significant others, such as family,

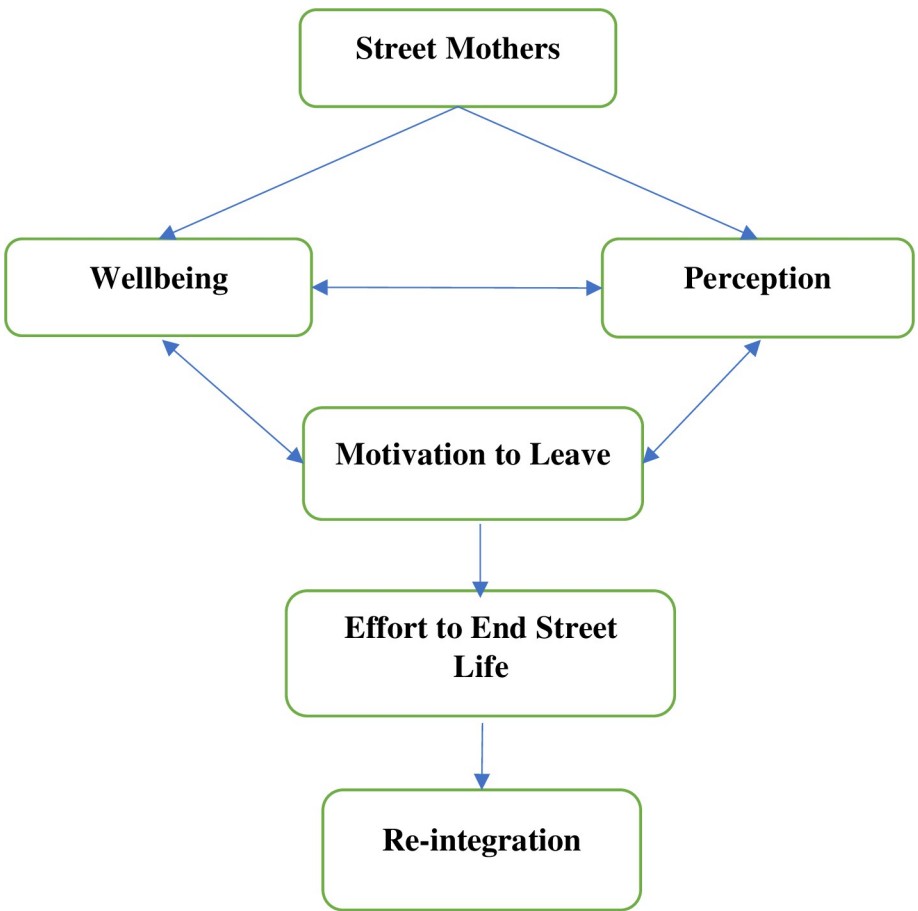

**Fig 1. A diagram showing the relationship between themes of lived experiences of street mothers in Bahir Dar city, July 2021.**

relatives and friends, is majorly in a state of disintegration and there is a lack of development of deep, affective bonds [43].

The current study demonstrated that the mental well-being of street mothers was challenged by excessive worry, anxiety and stressful life. Parental communication is one of the key ways in which the family can act as a protective health asset, promoting pro-social values that equip young people to deal with stressful situations or buffer them against adverse influences [40]. Similarly, our study explored that fear of child car accident, child theft and violence, and children's future life affected the emotional well-being of street mothers.

The Universal Declaration of Human Rights acknowledges the importance of everyone's access to food, clothing, housing and medical care for adequate health and well-being of oneself and their family [41]. But in this study insecure basic needs and insecure daily income were the main challenges of the economic well-being of street mothers. Literature also remarked that street beggars are vulnerable group with health challenges secondary to difficulty of accessing health services because of physical disabilities and low socioeconomic status [44]. In other words, lack of financial resources and negative judgements were barriers to well-being [45].

Ghaderi 2018 noted that most experts recognized human connection with God as the most important part of spiritual health or well-being. The connection with God is characterized by

knowing God, feeling affection and love toward God, laying one's hopes on God, being thankful for divine blessings, and prayer [46]. In the current study spiritual well-being of street mothers was demonstrated by their wordings and practice. The main spiritual expressions of street mothers were praying, thanking God, attending holy water, and taking flesh-blood and joining monastery life. Additionally, the spirituality of interviewed street mothers was demonstrated via mentioning the name of God in their wordings. This implies that street mothers showed strong spirituality or trust in God.

There is clear interaction between the well-being domains. The interconnection asserted by previous study revealed that neighbourhood that engender high levels of social capital create better mental health, more health-promoting behaviors, fewer risk-taking behaviors, better overall perceptions of health and greater likelihood of physical activity. Building neighbourhood social capital is therefore a means of tackling health inequalities [40]. There is study remarking that social support from a close relationship can mitigate the effects of poverty on mental well-being [45]. Another study shows that peer support contributed to reducing low mood and anxiety by overcoming feelings of isolation, disempowerment and stress, and increasing feelings of self-esteem, self-efficacy and parenting competence [47].

Environmental and social resources such as peace, economic security, a stable ecosystem, and safe housing, and individual resources such as healthful diet, social ties, resiliency, positive emotions, and autonomy are fundamental for well-being [48], such things were not met in the current study participants. As a result, there was no single participant who fulfill the general description of well-being such as presence of positive emotions and moods (contentment, happiness), the absence of negative emotions (depression, anxiety), satisfaction with life, fulfillment and positive functioning [8, 27, 49, 50].

The current study revealed that most street mothers had bad perception about street life. The street mothers have developed such perceptions because of their lived life challenges on street. Similarly, a study done in Zimbabwe revealed that street vending mothers generally had negative perceptions regarding concurrent childcare and street vending activities. Motherhood with street vending expose children to street vending environments that seriously undermining the children's well-being [16].

In the current study, the lived life challenges of street mothers were extended exposure to extreme weather conditions, lack of safe sleeping space at night, insults, increased vulnerability of their children to road traffic accidents and trafficking, and insecure basic needs. Similarly, another study underlined that concurrent street vending and childcare activities were concomitant with poor nurturance of children; elevated children's vulnerability to road traffic accidents, child trafficking, exposure to diseases, and prematurely socializing children to the love for money [16]. Street mothers' perception about children's vulnerability to accidents has strong foundation. A study asserts that injuries as inevitable events among children in the presence of hostile situations that place children at increased risk of injuries in settings such as lack of adult supervision, and lack of safe play areas. Close child supervision was highlighted as key in preventing injuries [51].

There were also some street mothers who felt that they were treated like normal people despite having health issues. The mothers might have their own grounds to have such perception. It might be due to the fact that some mothers actually received bad judgments from people while some others actually didn't experience bad judgments. A study done in Tanzania revealed that street children know they are called different names and they seem not to agree with those names. Thus, street children's identification of themselves comes from what other people identify them with, which may have implication for them not to want to go back to their homes or to school [52].

The current study explored that almost all street mothers had desire to leave street life with the intent to improve children's or their own life or both. The motivational factors include desire to educate and invest on their children, desire to self-contained life, desire to saving money, desire to have played job and envisioned better future work for their children. Another study also remarked that when street children had started to think about who they would be in the future and what they would do, it gave them an extra incitement to try to go to school and change their life. Overall, the wish to get a family and offering the future family a better life than what they had become a motivator to leave street life [53].

In the current study, beyond desire to leave street life, some street mothers have tried different mechanisms to end the street life. These include educating their children, giving their children to others, and let their children devote their time on work and education.

The study had limitations. The well-being of children was assessed by interviewing their mothers though they were few numbers, only one mother had a child whose age greater than 12 years. But, the best teachers for those children age above 12 years old are themselves. Additionally, there is no universally accepted definition of well-being. But we have used the WHO definition to guide the study and organize the sub-themes of well-being framework.

## Conclusion

Street mothers and their children had a very poor status in almost all dimensions of well-being. The perception of mother about street life was almost negative. The mothers had strong motivation to end street life but they were unable to make strong tangible efforts, which implies that they need assistance mechanisms before they change to street extended families under misery. Therefore, the findings call for concerted efforts from various stakeholders such as the Labor and Social Affairs of Amhara region, and Regional Health Bureau to change the life situation of street mothers in the study area. Moreover, the negative perception about street life and the strong motivation to end street life demonstrated in the current study must be used as an entry point to reintegrate street mothers to normal life situations.

## Supporting information

**S1 Appendix. In-depth interview guide (English and Amharic versions).**
(PDF)

**S2 Appendix. Thematic framework showing themes, subthemes and codes used during analysis.**
(PDF)

**S1 Dataset. Study dataset.**
(PDF)

## Acknowledgments

We are very grateful to Bahir Dar University for giving us ethical clearance. We are also indebted to thank the Labor and Social Affairs of Amhara regional state for offering us a support letter to pursue the study. Lastly, we acknowledge study participants for their cooperation during data collection.

## Author Contributions

**Conceptualization:** Dabere Nigatu, Gebeyehu Tsega, Shiferaw Birhanu, Yinager Workineh, Christian Tadele, Fentie Ambaw.

**Data curation:** Dabere Nigatu, Gebeyehu Tsega, Shiferaw Birhanu, Yinager Workineh, Christian Tadele.

**Formal analysis:** Dabere Nigatu, Gebeyehu Tsega, Shiferaw Birhanu, Yinager Workineh, Christian Tadele.

**Investigation:** Dabere Nigatu, Gebeyehu Tsega, Shiferaw Birhanu, Yinager Workineh.

**Methodology:** Dabere Nigatu, Gebeyehu Tsega, Shiferaw Birhanu, Yinager Workineh, Christian Tadele, Fentie Ambaw.

**Project administration:** Dabere Nigatu, Gebeyehu Tsega, Shiferaw Birhanu, Yinager Workineh, Christian Tadele.

**Resources:** Dabere Nigatu, Gebeyehu Tsega, Shiferaw Birhanu, Yinager Workineh, Christian Tadele.

**Supervision:** Fentie Ambaw.

**Validation:** Dabere Nigatu, Gebeyehu Tsega, Shiferaw Birhanu, Yinager Workineh, Fentie Ambaw.

**Writing – original draft:** Dabere Nigatu.

**Writing – review & editing:** Dabere Nigatu, Gebeyehu Tsega, Shiferaw Birhanu, Yinager Workineh, Christian Tadele, Fentie Ambaw.

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
