## [Decision Letter · Decision Letter 0]

26 Sep 2022

PONE-D-22-20787Street Mothers’ Well-Being and Motivation to Leave Street Life in Bahir Dar City, Ethiopia: A Phenomenological StudyPLOS ONE

Dear Dr. Nigatu,

Thank you for submitting your manuscript to PLOS ONE. After careful consideration, we feel that it has merit but does not fully meet PLOS ONE’s publication criteria as it currently stands. Therefore, we invite you to submit a revised version of the manuscript that addresses the points raised during the review process.

We look forward to receiving your revised manuscript.

Kind regards,

Benedict Weobong, Ph.D

Academic Editor

PLOS ONE

Journal Requirements:

Additional Editor Comments:

Kindly ensure all comments are addressed.

**1.**
**Main comments**

**Line 27-28**: What is perception of street mothers about street life? The results section of the abstract does not capture that. Shed a light on that.

**Line 65-66**: “This made to have many people lead their livelihood on street overtime.” This sentence does not make sense. Articulate properly the linkage between many Ethiopians living in poverty and streetism. 

**Line 79:** Rephrase sentence in line 79. The evidence provided from line 80 to 83 to support the claim that street people are on the rise is not clear. Before concluding that street people are on the rise there should be a comparison of data between previous year and current year. You could perhaps either look at data either before or after 2018 and compare with 2018. 

**Line 87-88:** The phrasing is a bit unclear. Should read: *For mothers with children, the lack of secure accommodation or creche facilities cannot allow **them** to move far from home to look for income opportunities.*

**
*Line 118-123:*
**
* provide rationale for choosing Bahir Dar city. What makes the site ideal for your study? Is the phenomenon of streetism prevalent in the city? Any figures to support that?*

**Line 137-139: **Perhaps you could break down the interview guide into themes. It could be in the form of a short table to give readers an insight of the kind of questions asked during the interview. It could also be an appendix.

***Line 131-133: ***Expatiate the process of the sampling selection. How did the authors select respondents. Did authors use snowball? Detail the steps used to select mothers from different age groups. Provide *enough information to validate the study and make it easy for another researcher to reproduce the study with the same methods. *

**Line 180-184: **I suggest the thematic framework developed during analysis be added as an appendix if possible. 

**Line 218-229:** These quotes do not necessarily represent the theme of physical wellbeing of street mothers highlighted. For instance, how does the physical problems of their husbands/children impact street mothers. If does impact them, would it not impact their mental wellbeing rather than physical wellbeing? 

**Line 245-247:** Is streetism a causal factor of this sentiment by family members or because of another factor

**Line 252-256: **I struggle to understand how the quotes provided is related to a bad relation with family and therefore affecting their social wellbeing. 

**Line 311-345: **I will suggest that emotional wellbeing and mental health are combined into one grand theme. The quotes/sub-themes under the two themes are almost the same and will be better to have them under one theme.

**Line 380:** Provide a context of what the holy water is. It would be good to enlighten your readers what ‘attending holy water’ means.

**Line 553-Line 556:** There seems to be a disconnect between line 553 to line 555 and the second part of line 555 to line 559. How are the broader social conditions linked to social well-being of your respondents? Need to make it clear

Line 576-577: Did the study cited from lines 576-577 establish a link between human connection with God and well-being? Perhaps you could discuss the study cited from that angle and link it to your current study on spiritual well-being.

**
*2.*
**
**
*Minor comments*
**

**
*Line 129-130:*
**
* “The researchers were also made discussion each day after interview.” Check grammar*

**Line 157-158: **“*Data were read and re-read by the research team until understanding the full content of the response.” **Check grammar*

**Line 192-195: **Check grammar and tenses

**Line 200: “***The age of street mothers participated in the study rage from 20 to 42 years.” *“rage” should be ranges

**Line 222**: “*Street mothers also stated as their children had physical health problem.” *Check tenses

**Line 380:** “holly water” Check spellings

**Line 514-516: “***Some street mothers have tried different mechanisms to end the street life such as educating their children, giving their children to their relatives and else persons, and let them devote their time on work and education.” *Check spellings and tenses

Line 521-522: “*Another a **40 years old mother described how she **have tried her best to educate her child and* *how she was challenged by her child.” *Check spellings and tenses

Reviewers' comments:

Reviewer's Responses to Questions

**Comments to the Author**

1. Is the manuscript technically sound, and do the data support the conclusions?

Reviewer #1: Yes

2. Has the statistical analysis been performed appropriately and rigorously? 

Reviewer #1: N/A

3. Have the authors made all data underlying the findings in their manuscript fully available?

Reviewer #1: Yes

4. Is the manuscript presented in an intelligible fashion and written in standard English?

Reviewer #1: Yes

5. Review Comments to the Author

Reviewer #1: The manuscript is technically sound and answers the research questions the authors sought to explore. However, authors need to address concerns raised regarding analysis and quotes provided in the results section.

6. PLOS authors have the option to publish the peer review history of their article (what does this mean?). If published, this will include your full peer review and any attached files.

Reviewer #1: No

---

## [Author Response · Author response to Decision Letter 0]

10 Nov 2022

Response to Academic Editor

Comment#1: Please ensure that your manuscript meets PLOS ONE's style requirements, including those for file naming.

Response#1: We have checked the alignment of the revised submission to PLOS ONE’s style requirements. 

Comment#2: In your Data Availability statement, you have not specified where the minimal data set underlying the results described in your manuscript can be found. PLOS defines a study's minimal data set as the underlying data used to reach the conclusions drawn in the manuscript and any additional data required to replicate the reported study findings in their entirety. All PLOS journals require that the minimal data set be made fully available. For more information about our data policy, please see http://journals.plos.org/plosone/s/data-availability.

Response#2: In the revised submission, we uploaded the study’s minimal underlying data set as Supporting Information files. 

Comment: 3. Please review your reference list to ensure that it is complete and correct. If you have cited papers that have been retracted, please include the rationale for doing so in the manuscript text, or remove these references and replace them with relevant current references. Any changes to the reference list should be mentioned in the rebuttal letter that accompanies your revised manuscript. If you need to cite a retracted article, indicate the article’s retracted status in the References list and also include a citation and full reference for the retraction notice.

Response#3: Thank you for the comment. We reviewed that the reference list given in the manuscript are correct and complete. 

Response to reviewer

Response to reviewer #1

General comment: This paper explores the well-being of street mothers and their children in Bahir Dar City, Ethiopia. It addresses an important topic, especially given that the perspectives of street mothers is significant input for policy makers and implementers working to improve the well-being of the vulnerable in Ethiopia. Using the phenomenological approach as a method, the authors found out that street mothers have poor wellbeing status in almost all different domains of wellbeing the study sought to explore. The study outlined life challenges such as lack of safe sleeping space at night, increased vulnerability of their children to road traffic, lack of access to basic needs were all contributing factors of poor wellbeing among street mothers. The authors added that despite the challenges faced by the street mothers, the desire to educate and invest in their children, desire to live a satisfying were among the motivational pushing them. The authors concluded that the mothers are unable to achieve these goals despite their efforts, hence the need for an assistance mechanism by stakeholders such as Labour and Social Affairs of Amhara region and Regional Health Bureau to support them. 

The writing is clear and structure of the paper satisfying. 

The main strength of this paper is that it addresses wellbeing of street mothers in holistic way. It gives an insight into the different components of wellbeing and their interconnectedness. As such this article represents a good study which has the potential influence the thinking of policy makers about addressing the wellbeing of the vulnerable holistically. Some of the weaknesses is that the study does not provide a thorough explanation of the various domains of wellbeing explored in the study. The linkages between the challenges faced by the respondents and the various domains of wellbeing.

Response: First, we are very grateful for your devotion and in-depth review of the paper and for giving us a clear, precise and comprehensive summary of what the paper is about. We have revised the manuscript by considering all the comments and suggestion given by reviewer and academic editor and produced concise and readable manuscript. 

Comment #1: Line 27-28: What is perception of street mothers about street life? The results section of the abstract does not capture that. Shed a light on that.

Response #1: Thank you for the insight. We have incorporated the comment in the revised submission. 

Comment #2: Line 65-66: “This made to have many people lead their livelihood on street overtime.” This sentence does not make sense. Articulate properly the linkage between many Ethiopians living in poverty and streetism.

Response #2: The authors revised the contents as per the comment. The authors’ intention was to express the link between poverty and streetism. Poverty is a predisposing factor for streetism. 

Comment #3: Line 79: Rephrase sentence in line 79. The evidence provided from line 80 to 83 to support the claim that street people are on the rise is not clear. Before concluding that street people are on the rise there should be a comparison of data between previous year and current year. You could perhaps either look at data either before or after 2018 and compare with 2018.

Response #3: Thank you for your comment, the authors agree that the concerns raised are robust. Actually, getting reliable and valid data on the number of homeless or street people in a given area or country is a challenge even in nations with robust census or civil registration systems. Enumeration of homeless or street individual is difficult because they typically seek to conceal themselves from the authorities. Additionally, the geographical extent to cover may encompass all the legal jurisdiction of a given country since they lack fixed residential place. In country like Ethiopia getting accurate data on the magnitude of street people and its trend is a critical challenge. Except these challenges the problem exists. Despite these challenges, still there are anecdotal evidences, qualitative expressions and some quantitative evidences showing the raise of streetism overtime. Even the current internal conflict situation in the country is assumed to exacerbate the problem now and then. The authors clearly addressed the problem in the revised submission.

Comment #5: Line 87-88: The phrasing is a bit unclear. Should read: For mothers with children, the lack of secure accommodation or creche facilities cannot allow them to move far from home to look for income opportunities.

Response #5: Thank for the comment and suggestion. We have revised as suggested by the reviewer.

Comment #6: Line 118-123: provide rationale for choosing Bahir Dar city. What makes the site ideal for your study? Is the phenomenon of streetism prevalent in the city? Any figures to support that?

Response #6: Thank you for the comment. Actually, we tried to provide the rationale for choosing Bahir Dar in the introduction section of the previous submission. Bahir Dar city is selected because the city has many street people who are primarily coming from rural settings. The reasons to have many street people in city are: being the regional capital, promising city for job, a popular tourist destination, big center for Orthodox Christian faith and, recently, it is one of non-conflict affected city of the Amhara region. We considered the comment in the revised manuscript.

Comment #7: … Line 137-139: Perhaps you could break down the interview guide into themes. It could be in the form of a short table to give readers an insight of the kind of questions asked during the interview. It could also be an appendix.

Response #7: We have incorporated the comment in the revised submission.

Comment #8: Line 131-133: Expatiate the process of the sampling selection. How did the authors select respondents? Did authors use snowball? Detail the steps used to select mothers from different age groups. Provide enough information to validate the study and make it easy for another researcher to reproduce the study with the same methods.

Response #8: Thank you for the comment and we considered presenting the detail of sample selection in the revised manuscript. First, we scanned the city for mapping of potential sites. During the scanning phase, we travelled on the main roads in the city to identify and have a mental map of the potential sites, where we could access street mothers for interview. Second, the interviewers visited the identified potential sites and interviewed the mothers. In addition to the main road sites, we visited the church areas since they are potential sites to access street mothers. During data collection, we had a common session to discuss participants mix by age and duration of stay on street. The discussion continued until we had a minimum adequate sample size and had reached the point of information saturation. 

Comment #9: Line 180-184: I suggest the thematic framework developed during analysis be added as an appendix if possible.

Response #9: We appended the thematic framework developed during analysis

Comment #10: Line 218-229: These quotes do not necessarily represent the theme of physical wellbeing of street mothers highlighted. For instance, how does the physical problems of their husbands/children impact street mothers. If does impact them, would it not impact their mental wellbeing rather than physical wellbeing?

Response #10: Thank you for the comment. Though the main aim of this study is to explore street mother’s well-being, we have assessed their child/ren’s well-being at the same time. The well-being of child/ren is assessed as perceived by the mother. For mothers, the well-being their children is very critical concern. The study also explored that there were women who were spending on street because of having physically disabled child. The mothers clearly marked that they were unable work as housemaid even because there were no one to care for their disable child. Hence, we kept the quotation explaining about their child physical problems in the manuscript while we omitted the quates explaining about husband’s physical problems. 

Comment #11: Line 245-247: Is streetism a causal factor of this sentiment by family members or because of another factor

Response #11: Yes, streetism is a causal factor for her experience. This street mother is explaining the ignorance she is receiving from her family because of streetism. She is expressing her grievance about the stigma and discrimination she endured because of living on street. Generally, she had disconnected relationship with her family. 

Comment #12: Line 252-256: I struggle to understand how the quotes provided is related to a bad relation with family and therefore affecting their social wellbeing.

Response #12: The quote given is about the respondent’s relationship with the oldest daughter. The oldest daughter is big in size and she usually beats her mother. Even, there are time her daughter tried to hang her mother. The authors brought this quotation to witness/show the relationship that the mother had with her daughter. The daughter quitted her schooling without her mother consent. The eldest daughter had bad relationship with her mother. 

Comment #13: Line 311-345: I will suggest that emotional wellbeing and mental health are combined into one grand theme. The quotes/sub-themes under the two themes are almost the same and will be better to have them under one theme.

Response #13: Thank you for your suggestion. We have merged them in the revised submission. 

Comment #14: Line 380: Provide a context of what the holy water is. It would be good to enlighten your readers what ‘attending holy water’ means.

Response #14: Thank you for your insight. We have given the context and meaning of “holy water” and “attending holy water” in the revised submission. According to Ethiopian Orthodox Christian faith, there are holy water sites known for healing miracles. The Orthodox Christian followers travel for miles or even days to seek the curative powers of holy water blessed by Orthodox priests. They drink holy water and/or submerge themselves in pools to receive healing. The frequency and duration of such practice can be determined by the managing priest and may vary from follower to follower. Overall, Orthodox Christian followers who are attending holy water sites believe that holy water has curative power and can heal any illness.

Comment #15: Line 553-Line 556: There seems to be a disconnect between line 553 to line 555 and the second part of line 555 to line 559. How are the broader social conditions linked to social well-being of your respondents? Need to make it clear

Response #15: Thank you, we have considered the comment in the revised submission.

Comment #16: Line 576-577: Did the study cited from lines 576-577 establish a link between human connection with God and well-being? Perhaps you could discuss the study cited from that angle and link it to your current study on spiritual well-being.

Response #16: We have revised the manuscript as per the comment.

Comment #17: Line 129-130: “The researchers were also made discussion each day after interview.” Check grammar

Response #17: Thank you. We have done the revision as per the comment.

Comment #18: Line 157-158: “Data were read and re-read by the research team until understanding the full content of the response.” Check grammar

Response #18: Thank you. We have done the revision as per the comment.

Comment #19: Line 192-195: Check grammar and tenses

Response #19: Thank you. We have revised as per the comment.

Comment #20: Line 200: “The age of street mothers participated in the study rage from 20 to 42 years.” “rage” should be ranges

Response #20: Thank you. We have revised as per the comment.

Comment #21: Line 222: “Street mothers also stated as their children had physical health problem.” Check tenses

Response #21: Thank you. We have revised as per the comment.

Comment #22: Line 380: “holly water” Check spellings

Response #22: Thank you. We have revised as per the comment.

Comment #23: Line 514-516: “Some street mothers have tried different mechanisms to end the street life such as educating their children, giving their children to their relatives and else persons, and let them devote their time on work and education.” Check spellings and tenses

Response #23: Thank you. We have revised as per the comment.

Comment #24: Line 521-522: “Another a 40 years old mother described how she have tried her best to educate her child and how she was challenged by her child.” Check spellings and tenses

Response #24: Thank you. We have revised as per the comment.

---

## [Editor Report · Decision Letter 1]

21 Nov 2022

Street Mothers’ Well-Being and Motivation to Leave Street Life in Bahir Dar City, Ethiopia: A Phenomenological Study

PONE-D-22-20787R1

Dear Dr. Nigatu,

We’re pleased to inform you that your manuscript has been judged scientifically suitable for publication and will be formally accepted for publication once it meets all outstanding technical requirements.

Kind regards,

Benedict Weobong, Ph.D

Academic Editor

PLOS ONE
---

## [Editor Report · Acceptance letter]

6 Dec 2022

PONE-D-22-20787R1 

Street Mothers’ Well-Being and Motivation to Leave Street Life in Bahir Dar City, Ethiopia: A Phenomenological Study 

Dear Dr. Nigatu:

I'm pleased to inform you that your manuscript has been deemed suitable for publication in PLOS ONE. Congratulations! Your manuscript is now with our production department. 

Kind regards, 

on behalf of

Dr. Benedict Weobong 

Academic Editor

PLOS ONE